# Nalfurafine Hydrochloride, a κ-Opioid Receptor Agonist, Induces Melanophagy via PKA Inhibition in B16F1 Cells

**DOI:** 10.3390/cells12010146

**Published:** 2022-12-29

**Authors:** Ha Jung Lee, Seong Hyun Kim, Yong Hwan Kim, So Hyun Kim, Gyeong Seok Oh, Ji-Eun Bae, Joon Bum Kim, Na Yeon Park, Kyuhee Park, Eunbyul Yeom, Kwiwan Jeong, Pansoo Kim, Doo Sin Jo, Dong-Hyung Cho

**Affiliations:** 1BK21 FOUR KNU Creative BioResearch Group, School of Life Sciences, Kyungpook National University, Daegu 41566, Republic of Korea; 2Brain Science and Engineering Institute, Kyungpook National University, Daegu 41566, Republic of Korea; 3Bio-center, Gyeonggido Business & Science Accelerator, Gyeonggido, Suwon 16229, Republic of Korea; 4OGASIS Corp. 260, Changyong-daero, Yongtong-gu, Suwon 08826, Republic of Korea

**Keywords:** nalfurafine hydrochloride, autophagy, κ-opioid receptor, melanophagy, B16F1 cells

## Abstract

Selective autophagy controls cellular homeostasis by degrading unnecessary or damaged cellular components. Melanosomes are specialized organelles that regulate the biogenesis, storage, and transport of melanin in melanocytes. However, the mechanisms underlying melanosomal autophagy, known as the melanophagy pathway, are poorly understood. To better understand the mechanism of melanophagy, we screened an endocrine-hormone chemical library and identified nalfurafine hydrochlorides, a κ-opioid receptor agonist, as a potent inducer of melanophagy. Treatment with nalfurafine hydrochloride increased autophagy and reduced melanin content in alpha-melanocyte-stimulating hormone (α-MSH)-treated cells. Furthermore, inhibition of autophagy blocked melanosomal degradation and reversed the nalfurafine hydrochloride-induced decrease in melanin content in α-MSH-treated cells. Consistently, treatment with other κ-opioid receptor agonists, such as MCOPPB or mianserin, inhibited excessive melanin production but induced autophagy in B16F1 cells. Furthermore, nalfurafine hydrochloride inhibited protein kinase A (PKA) activation, which was notably restored by forskolin, a PKA activator. Additionally, forskolin treatment further suppressed melanosomal degradation as well as the anti-pigmentation activity of nalfurafine hydrochloride in α-MSH-treated cells. Collectively, our data suggest that stimulation of κ-opioid receptors induces melanophagy by inhibiting PKA activation in α-MSH-treated B16F1 cells.

## 1. Introduction

Autophagy is a self-degradative process that removes damaged or unnecessary organelles as well as misfolded or aggregated proteins [1]. Upon autophagy activation, the isolation membrane encloses a portion of the cytoplasm to form an autophagosome, which engulfs target components and subsequently fuses with the lysosome to form an autolysosome [2]. Autophagy-related genes (ATG) are essential for autophagy activation in the regulation of autophagosomes and autolysosome formation [3]. Two ubiquitin-like systems are involved in autophagic vesicle formation: ATG7, an E1-like activating enzyme, binds to ATG8/microtubule-associated protein light chain 3 (LC3) or ATG12 and is then transferred to one of the E2-like conjugation enzymes, ATG3 and ATG10. The ATG12-ATG5 complex then conjugates with ATG16, an E3-like enzyme for the ATG8-PE conjugate, binding to the autophagosome membrane through a lipidation reaction [4]. Although autophagy is considered a non-selective bulk-degradation process under starvation conditions, it can remove specific target organelles. For example, organelles such as mitochondria (mitophagy) and peroxisomes (pexophagy) can be eliminated by selective autophagy [5,6]. Transcription factor EB (TFEB) is a major transcriptional regulator of autophagy genes [7]. Under normal conditions, TFEB is retained in the cytoplasm after phosphorylation by the mammalian target of rapamycin (mTOR). However, activation of autophagy in response to different stimuli, including starvation or mTOR inhibition, leads to dephosphorylation of TFEB and its rapid translocation to the nucleus, resulting in the expression of its target genes [8,9].

The skin is the largest organ of the body, which serves as a protective barrier against various stress stimuli, such as UV exposure. Structurally, the epidermis and the outer layer of the skin are mainly composed of keratinocytes and melanocytes. The dermis, which is the inner layer of the skin, contains connective tissue and hair follicles. The subcutaneous layer consists of fat and provides the primary structural support to the skin [10]. Melanocytes are specialized melanin-producing cells found in the skin, hair follicles, eyes, and brain, originating from neural crest melanoblasts [11]. In the skin, melanocytes generate melanin pigment in the melanosome and transfer them to the surrounding keratinocytes [10]. A series of interactions collectively termed melanogenesis, then occur to synthesize melanin by catalysis with enzyme complexes. Several pigmentation disorders trigger skin discoloration, including melasma, albinism, and vitiligo [12]. Melanosome formation and maturation occur during melanogenesis, and several factors, such as tyrosinase and tyrosinase-related protein 1/2 (TRP1/2), control melanin production [13]. As a transcription factor, microphthalmia-associated transcription factor (MITF) mainly controls the expression of melanogenesis-related proteins, including tyrosinase and TRP1/2 [14]. α-melanocyte-stimulating hormone (α-MSH) transactivates MITF by increasing the cyclic adenosine monophosphate (cAMP)-cAMP response element-binding (CREB) signaling cascade [14,15]. MITF target genes are enriched in DNA replication and repair, mitotic events, and pigmentation [16,17].

Recently, our group demonstrated that autophagy regulates pigmentation in melanocytes by controlling melanosome-selective autophagy, melanophagy [18]. Hormones produced by glands can circulate throughout the body to trigger different effects in target cells, tissues, and organs [19]. However, the endocrinological regulation of autophagy and melanophagy remains unexplored. Therefore, we screened an endocrinology-hormone library in B16F1 melanoma cells to identify novel melanophagy regulators associated with hormones and identified nalfurafine hydrochloride as the most potent inducer of melanophagy. Opioid receptors are G-protein coupled receptors (GPCRs), of which nalfurafine hydrochloride is highly selective for κ-opioid receptors and has been approved for treating central pruritus in patients with liver disease [20]. However, the effect of nalfurafine hydrochloride on skin pigmentation has not been investigated. In this study, we observed that activation of the κ-opioid receptor with nalfurafine hydrochloride strongly inhibited pigmentation by promoting melanosomal autophagy (melanophagy) in B16F1 cells.

## 2. Materials and Methods

### 2.1. Reagents and Plasmids

Nalfurafine hydrochloride ((2E)-*N*-[(5α,6β)-17-(cyclopropylmethyl)-3,14-dihydroxy-4,5-epoxymorphinan-6-yl]-3-(3-furyl)-*N*-methylacrylamide hydrochloride), and Mianserin hydrochloride (1,2,3,4,10,14b-Hexahydro-2-methyl-dibenzo[c,f]pyrazino [1,2-a]azepine hydrochloride) were purchased from MedChemExpress (Monmouth Junction, NJ, USA). MCOPPB (1-[1-(1-methylcyclooctyl)-4-piperidinyl]-2-(3R)-3-piperidinyl-1H-benzimidazole, trihydrochloride, hydrate) was purchased from Caymanchem (Ann Arbor, MI, USA). α-melanocyte-stimulating hormone (α-MSH), bafilomycin A1, and forskolin were purchased from Sigma-Aldrich (St. Louis, MO, USA). ARP101 and Torin1 were purchased from TOCRIS (Bristol, UK). The expression plasmid pEGFP-LC3 was a gift from Tamotsu Yoshimori (Osaka University, Osaka, Japan). The plasmids pEGFP-TFEB (38119), pmRFP-EGFP-LC3 (21074), and pEGFP-two-pore channel (TPC2) (80153) were purchased from Addgene (Watertown, MA, USA). For pcDNA/TPC2-mRFP-EGFP plasmid construction, the PCR-amplified products TPC2 and mRFP-EGFP were individually subcloned into the pcDNA3.1/Myc-His(−)A vector. A validated small interfering RNA (siRNA) targeting mouse Atg5 (5′-ACCGGAAACUCAUGGAAUA-3′) and scrambled control (5′-CCUACGCCACCAAUUUCGU-3′) were synthesized by Genolution (Seoul, Korea).

### 2.2. Cell Culture

B16F1 melanoma cells, obtained from the American Type Culture Collection (Manassas, VA, USA), were cultured at 37 °C in a 5% CO_2_ incubator and maintained in Dulbecco’s modified Eagle’s medium containing 10% fetal bovine serum and 1% penicillin/streptomycin (Invitrogen, Carlsbad, CA, USA). To generate stable cell lines, B16F1 melanoma cells were transfected with pEGFP-LC3 (B16F1/GFP-LC3), pcDNA/TPC2-mRFP-EGFP (B16F1/TPC2-mRFP-EGFP), or pEGFP-TFEB (B16F1/TFEB) using Lipofectamine 2000, according to the manufacturer’s protocol (Invitrogen). Stable transfectants were selected by growth in a selection medium containing 1.25 mg/mL of G418 (Invitrogen) for 10 days, and colonies derived from single transfected cells were isolated. A fluorescence microscope (IX71; Olympus) was used to select stable clones.

### 2.3. Cell-Based Hormone Library Screening

For cell-based hormone library screening, an endocrinology-hormone library was purchased from TargetMol (L2400) (Boston, MA, USA). B16/GFP-LC3 cells were seeded in 96-well plates. After 24 h, 1, 10, and 100 μM of the hormone library was added to each well. The GFP-LC3 puncta in the cells were monitored using fluorescence microscopy. The experiments were repeated twice and yielded consistent results.

### 2.4. Melanin Content Assay

Melanin content was determined using a slightly modified version of a previously described method. To measure melanin content, B16F1 cells were harvested by trypsinization and dissolved in a solubilization buffer at 100 °C for 30 min. The relative melanin content was determined by measuring the absorbance at 405 nm using a microplate reader (BioTek, Santa Clara, CA, USA).

### 2.5. Autophagy Analysis and Melanophagy Assay with Fluorescent Punctuation

For the autophagy assay, B16F1/GFP-LC3 cells were treated with nalfurafine hydrochloride (100 μM) or ARP 101 (10 μM). Autophagy was determined by the number of cells with GFP-LC3 punctate structures, indicative of autophagosomes, via fluorescence microscopy (IX71, Olympus, Tokyo, Japan). For the melanophagy assay, B16F1/TPC2-mRFP-EGFP cells were seeded onto coverslips in 12-well plates. The cells were pre-treated with α-MSH (0.5 μM) for 48 h and then incubated with nalfurafine hydrochloride (100 μM) in the presence or absence of bafilomycin A1 (100 nM) for 24 h. Subsequently, cells were washed with phosphate-buffered saline (PBS, pH 7.4), fixed with 4% paraformaldehyde at room temperature for 20 min, and then washed with PBS. After mounting on coverslips, cells were evaluated under a confocal microscope (LSM 800; Objective C-Apochromat 40×/1.2 W Corr UV-VIS-IR M27; Carl Zeiss, Thornwood, NY, USA). The number of cells with red punctate structures was counted, and the findings are presented as a percentage of the total counts of 200 cells.

### 2.6. Western Blotting

All lysates were prepared using 2 × Laemmli sample buffer (Bio-Rad, Hercules, CA, USA). Total protein was measured using the Bradford assay (Bio-Rad), according to the manufacturer’s instructions. The samples were separated using SDS-polyacrylamide gel electrophoresis (PAGE) and transferred to polyvinylidene fluoride (PVDF) membranes. After blocking with 4% skim milk in Tris-buffered saline supplemented with Tween-20, the membranes were incubated with primary antibodies, including anti-LC3 (NB100-2220), anti-ATG5 (NB110-53818, NOVUS Biologicals, Littleton, CO, USA), anti-TYR (sc-20035), anti-ABCD3 (sc-514728), anti-TOMM20 (sc-17764) and anti-P4HB (sc-20132; Santa Cruz Biotechnology, Dallas, TX, USA), anti-FTCD (ab27043; Abcam, Cambridge, UK), anti-phospho-p70S6K (9205), anti-p70S6K (9202), anti-phospho-TFEB (37681), anti-TFEB (4240), anti-phospho-PKA (4781) and anti-PKA (4782; Cell Signaling Technology, Danvers, MA, USA). Additionally, anti-ACTA1 (MAB1501, Sigma Aldrich, St. Louis, MO, USA). For protein detection, membranes were incubated with HRP-conjugated secondary antibodies (Pierce, Rockford, IL, USA). Additionally, the protein levels were further analyzed by a CS analyzer software (ATTO, Tokyo, Japan).

### 2.7. Statistical Analysis

Data were obtained from at least three independent experiments and are presented as the mean ± SEM. Statistical evaluation of the results was performed using one-way ANOVA. Data were considered significant at *p* < 0.05 (*), *p* < 0.01 (**), *p* < 0.001 (***).

## 3. Results

### 3.1. Nalfurafine Hydrochloride Induces Autophagy Activation in B16F1 Cells

Autophagy is an important quality control system in skin aging. To identify novel autophagy regulators in the skin, we established a stable cell line with GFP-LC3 in B16F1 cells (B16F1/GFP-LC3) [18] and performed a cell-based high-content screening with an endocrinology-hormone library. From this screening, we identified nalfurafine hydrochlorides as a potent inducer of autophagy in B16F1 cells. To validate the screening results, B16F1/GFP-LC3 cells were treated with either nalfurafine hydrochloride or ARP101, a potent inducer of autophagy [21]. As shown in Figure 1A, we confirmed that the formation of punctate GFP-LC3 protein substantially increased in nalfurafine hydrochloride-treated cells (Figure 1A). To further address the increased autophagic flux induced by nalfurafine hydrochloride, B16F1 cells were treated with bafilomycin A1. The protein level of LC3-II was more accumulated in cells treated with nalfurafine hydrochloride and bafilomycin A1 than that in control cells. These results indicate that nalfurafine hydrochloride is a potent autophagy inducer in B16F1 cells (Figure 1B). Transcription factor EB (TFEB) is a major regulator of autophagy and lysosomal biogenesis. Thus, we observed that treatment with nalfurafine hydrochloride induced nuclear translocation of TFEB in B16F1 cells (Figure 1C) [22]. Torin1, a potent mTOR inhibitor was used as a positive control. The protein level of TFEB phosphorylation was increased at the basal level; however, treatment with nalfurafine hydrochloride or Torin1 induced dephosphorylation of TFEB in B16F1 cells (Figure 1D). Concordantly, treatment with nalfurafine hydrochloride decreased the phosphorylation of p70S6K in α-MSH-treated B16F1 cells (Figure 1E), suggesting that nalfurafine hydrochloride activates autophagy by inhibiting mTOR activation.

### 3.2. Nalfurafine Hydrochloride Promotes Melanosomal Degradation by Inducing Melanophagy

Our group recently reported that the induction of autophagy controls melanin content [18,23]. To examine the whitening effect of nalfurafine hydrochloride, B16F1 cells stimulated with α-MSH were incubated with nalfurafine hydrochloride or arbutin, a potent anti-melanogenic agent [24]. Consistently, despite the strong melanogenic stimulus induced by α-MSH, nalfurafine hydrochloride significantly reduced the melanin content in B16F1 cells (Figure 2A). Cellular organelles can be degraded by selective autophagy [25]. As we observed that nalfurafine hydrochloride decreased melanin content and induced autophagy in B16F1 cells, we further examined the effect of nalfurafine hydrochloride on melanophagy. To confirm these results, we developed a melanophagy monitoring system using TPC2, a melanosome-membrane protein, followed by tandem fluorescent tags (mRFP-EGFP). Similar to the mRFP-EGFP-LC3 protein for autophagy flux assay, the basic principle of the tandem assay involves the difference in pH sensitivity of the red (mRFP) and green (EGFP) fluorescent proteins [26].

During melanophagy, targeted melanosomes are enclosed by autophagosomes, which are subsequently transported to lysosomes. In lysosomes, the green fluorescence signal is readily quenched as GFP is more acid-sensitive than RFP, whereas the red signal remains stable, suggesting melanophagy. Based on this novel monitoring system, B16F1/TPC2-mRFP-EGFP cells were treated with nalfurafine hydrochloride in the presence or absence of bafilomycin A1. As shown in Figure 2B, nalfurafine hydrochloride treatment increased the number of RFP-positive dots, which were blocked by bafilomycin A1 (Figure 2B). To investigate melanosome-selective autophagy, we further examined other organelles, including the mitochondria, ER, Golgi, and peroxisomes, in nalfurafine hydrochloride-treated cells. Consistently, melanosomal proteins such as tyrosinase were degraded, however, other membrane proteins, including a mitochondrial protein (TOMM20), ER protein (P4HB), Golgi protein (FTCD), and peroxisomal protein (ABCD3), were not substantially altered in nalfurafine hydrochloride-treated cells (Figure 2C), suggesting that nalfurafine hydrochloride induces melanosomal degradation.

Next, we investigated the effects of autophagy inhibition on nalfurafine hydrochloride-induced melanophagy. ATG5 is an essential autophagy regulatory protein involved in the extension of the phagocytic membrane in the autophagosome. Thus, the loss of ATG5 almost completely blocks autophagy activation [27]. Notably, the knockdown of Atg5 suppressed the nalfurafine hydrochloride-induced decrease in melanin content (Figure 3A). Moreover, the depletion of Atg5 restored the reduced levels of tyrosinase in nalfurafine hydrochloride-treated cells (Figure 3B). In addition, inhibition of autophagic flux by bafilomycin A1 also restored decreased melanin content by nalfurafine hydrochloride (Figure 3C). Collectively, these results further suggest that nalfurafine hydrochloride induces melanosomal degradation by promoting melanophagy in B16F1 cells.

### 3.3. Activation of the κ-Opioid Receptor Induces Melanophagy in B16F1 Cells

Nalfurafine hydrochloride is a selective kappa (κ)-opioid receptor agonist [28]. Therefore, we examined the effect of activation of κ-opioid receptors by other potent κ-opioid receptor agonists such as MCOPPB and mianserin on melanophagy in B16F1 cells. Similar to nalfurafine hydrochloride, MCOPPB or mianserin also strongly induced autophagic puncta with GFP-LC3 and accumulation of LC3 II in B16F1 cells (Figure 4A,B). Consistently both MCOPPB and mianserin significantly inhibited the excessive melanin content in α-MSH-stimulated B16F1 cells (Figure 4C). These results suggested that stimulation of the κ-opioid receptor influences melanophagy activation in B16F1 cells.

### 3.4. Inhibition of PKA Mediates Melanophagy in Nalfurafine Hydrochloride-Treated Cells

Opioid receptors are widely expressed throughout the nervous system. Thus, their physiological roles have been intensively elucidated in nervous systems [29,30]. However, the association between κ-opioid receptors and skin melanogenesis has not been explored. Therefore, we further investigated the potential regulatory mechanism of κ-opioid receptor-mediated melanophagy. It was reported that activation of the κ-opioid receptor inhibits the cyclic adenosine monophosphate (cAMP)/protein kinase A (PKA) signaling pathway [31,32]. Consistent with this notion, we also observed that treatment with nalfurafine hydrochloride inhibited PKA phosphorylation in α-MSH-stimulated B16F1 cells (Figure 5A). However, it was restored by treatment with forskolin, which directly increased intracellular cAMP levels by activating adenylyl cyclase (Figure 5A) [33]. Previously, we found that nalfurafine hydrochloride reduces the phosphorylation of p70S6K, which is a downstream target for mTOR signaling (Figure 1C). Therefore, we investigated the role of cAMP in nalfurafine hydrochloride-induced mTOR inhibition. As shown in Figure 5A, treatment with forskolin recovered the deceased phosphorylation of p70S6K, suggesting that activation of cAMP/PKA inhibits autophagy by modulating the mTOR signaling pathway (Figure 5A).

Therefore, we examined the inhibitory effect of PKA on melanophagy in nalfurafine hydrochloride-treated cells. The enhancement of GFP-LC3 puncta by nalfurafine hydrochloride was significantly reduced by combination treatment with forskolin in B16F1/GFP-LC3 cells (Figure 5B). Furthermore, treatment with forskolin significantly inhibited the autophagic degradation of melanosomes by nalfurafine hydrochloride in B16F1/TPC2-mRFP-EGFP cells (Figure 5C). Consistent with these results, we found that nalfurafine hydrochloride treatment reduced melanin content in α-MSH-stimulated B16F1 cells while forskolin restored the reduced melanin content in nalfurafine hydrochloride-treated cells (Figure 5D). Collectively, our results suggest that κ-opioid receptor agonists induce melanophagy by inhibiting PKA activation in B16F1 cells.

## 4. Discussion

As melanin is the primary determinant of mammalian skin pigmentation, disorders in melanin production and melanosome transport to keratinocytes are associated with various pigmentary diseases, such as melasma, vitiligo, and ash leaf spots [34]. Although the quality and quantity control of melanosomes by melanophagy are vital mechanisms for understanding pigmentary diseases, the precise regulatory events underlying melanophagy remain largely unknown. Hormones are strongly implicated in the maintenance of skin homeostasis, thus disturbances in hormonal regulation are involved in various skin perturbations [35]. In this study, we screened an endocrine hormone library and identified several candidates for autophagy inducers such as moxisylyte hydrochloride, tamsulosin hydrochloride, balicatib, VTP27999, aminoglutethimide, and metyrapone as well as nalfurafine hydrochloride. Notably, it was previously reported that aminoglutethimide and metyrapone induce autophagy in different cells [36,37]. In this study, a newly developed monitoring system (B16F1/TPC2-mRFP-EGFP cells) for melanosomal degradation revealed that nalfurafine hydrochloride, a selective agonist of the κ-opioid receptor, induces melanophagy (Figure 2B). Nalfurafine hydrochloride has previously been clinically used for treating itching in patients undergoing kidney dialysis and those with chronic liver diseases [38]. Several opioid receptors contribute to numerous physiological processes, including pain control, reproduction, growth, respiration, and immune reactions [39]. Among them, the κ-opioid receptor is predominantly expressed in the central nervous system, but it is also expressed in the adrenal medulla, digestive tissues, heart, kidney, placenta, peripheral vasculature, uterus, and immune cells [30,40]. Notably, the κ-opioid receptor is upregulated in several solid tumors and is associated with cancer development and poor prognosis, and it mediates the immunosuppressive effects [41,42]. Despite the multiple functions of the κ-opioid receptor in various tissues, its role in skin pigmentation has not yet been elucidated. In this study, we elucidated the effect of the κ-opioid receptor activation on the regulation of skin pigmentation by addressing an effect of nalfurafine hydrochloride on melanosomal degradation. Blockage of autophagy by Atg5 knockdown or bafilomycin A1 substantially restored the reduced melanin content and inhibited the autophagy by nalfurafine hydrochloride in α-MSH-stimulated B16F1 cells (Figure 3). Our findings support the hypothesis that stimulation of the κ-opioid receptor with nalfurafine hydrochloride decreases melanin content by activating melanophagy.

Hyperactivation of the κ-opioid receptor with dynorphin, an endogenous opioid peptide in mouse hippocampal neurons, exerts an anti-epileptic effect by activating the mTOR signaling pathway, which is a major autophagy regulatory pathway [43]. In addition, stimulation of the κ-opioid receptor by the chemical agonist, U50488H, protects against hypoxic pulmonary hypertension by inhibiting autophagy via adenosine monophosphate-activated protein kinase (AMPK)-mTOR signaling [44]. These reports suggest that activation of the κ-opioid receptor inhibits autophagy by activating the mTOR pathway. Nonetheless, we found that treatment with nalfurafine hydrochloride did not activate but inhibit the mTOR pathway in B16F1 melanoma cells (Figure 1C–E). TFEB activity is largely controlled by its subcellular localization. Phosphorylated TFEB is sequestered into the cytosol; hence, transcriptional induction of its target genes is inhibited. In contrast, dephosphorylated TFEB rapidly translocates to the nucleus to promote the expression of its target genes [7]. Similar to Torin1, treatment with nalfurafine hydrochloride induced the translocation of TFEB by inhibiting mTOR signaling in B16F1 cells (Figure 1C,D). These results suggest an alternative mechanism of nalfurafine hydrochloride in autophagy activation via regulation of mTOR signaling in B16F1 cells. Therefore, we explored another potential mechanism for nalfurafine hydrochloride-mediated melanophagy. Opioid receptors are G protein-coupled receptors (GPCRs) that mediate multiple intracellular signaling pathways by modulating cAMP and calcium [30,45,46]. Thus, stimulation of the κ-opioid receptor activates signaling kinase cascades, including G protein-coupled receptor kinases and mitogen-activated protein kinases (MAPK) proteins [30,46]. Some opioid receptors transduce signals through G inhibitory proteins (G_i_) to inhibit adenylyl cyclase, subsequently decreasing cAMP production and inactivating PKA. Concordantly, activation of the κ-opioid receptor with nalfurafine hydrochloride/TRK820 inhibits the cAMP/PKA signaling pathway to suppress vascular endothelial growth factor receptor 2 (VEGFR2) expression in endothelial cells [31,32]. Furthermore, treatment with TRK820 sufficiently blocked tumor development and angiogenesis in a xenograft mouse model [31]. In contrast, treatment with a μ-opioid receptor agonist, DAMGO, or a δ-opioid receptor agonist, SNC80, did not prevent angiogenesis in human umbilical vein endothelial cells [31]. These reports suggest that κ-opioid receptors suppress angiogenesis by inhibiting cAMP/PKA signaling. Notably, treatment with α-MSH stimulated the melanocortin 1 receptor (MC1R) to activate adenyl cyclase, which induced an increase in cAMP levels. Thus, cAMP-inducing agents, such as forskolin, lead to increased melanin content and the expression of melanin-producing proteins such as tyrosinase [47,48]. However, the effect of cAMP on melanophagy has not been elucidated. It was reported that cAMP signaling pathway is linked to AMPK activation, which is key regulatory protein for mTOR signaling. For example, elevated cAMP promotes AMPK phosphorylation at Thr 172, which subsequently promotes autophagy by inhibiting mTOR signaling [49]. In this study, we confirmed that nalfurafine hydrochloride inhibited the phosphorylation of PKA and p70S6K in α-MSH-treated B16F1 cells (Figure 5A). However, combination treatment with forskolin and nalfurafine hydrochloride largely reversed the decreased phosphorylation of PKA and p70S6K as well as melanin content induced in α-MSH-treated cells (Figure 5A,D). Notably, forskolin inhibited nalfurafine hydrochloride-induced melanophagy in α-MSH-treated B16F1 cells (Figure 5C). A recent study reported that cAMP might inhibit or promote autophagy depending on the cell type [50]. The cAMP/PKA signaling cascade is compartmentalized in distinct functional units termed microdomains [51]. Our findings suggest that inhibition of cAMP/PKA signaling promotes the autophagy-dependent clearance of melanosomes in B16F1 cells. Thus, further studies on downstream pathways, including CREB and MAPK proteins, and transcriptional control with MITF and TFEB will help understand the underlying mechanism of melanosome degradation in nalfurafine hydrochloride-treated cells.

## 5. Conclusions

In conclusion, our findings suggest that activation of the κ-opioid receptor by nalfurafine hydrochloride promotes melanophagy in α-MSH-treated melanocytes. Thus, we provided novel insights into the underlying mechanism of melanophagy and highlighted the potential of nalfurafine hydrochloride to be used as an ingredient in skin-care cosmetics.

## Figures and Tables

**Figure 1 cells-12-00146-f001:**
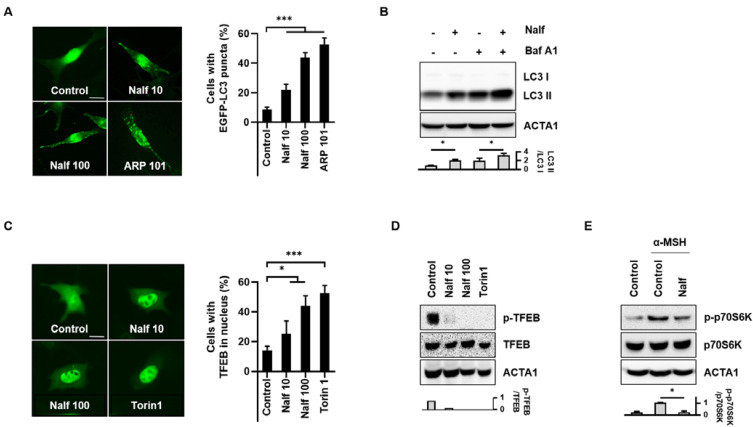
Nalfurafine hydrochloride induces autophagy by inducing translocation of TFEB in α-MSH-treated B16F1 cells. (**A**) B16F1/GFP-LC3 cells were incubated with either nalfurafine hydrochloride (Nalf, 10, 100 μM) or ARP 101 (10 μM). (**A**) After 24 h treatment, the cells were fixed and imaged with GFP fluorescence. Cells in which autophagy was activated were determined by counting punctate GFP-LC3 dots under a fluorescence microscope. (**B**) B16F1 cells were treated with nalfurafine hydrochloride (Nalf, 100 μM) in the presence or absence of bafilomycin A1 (Baf A1, 100 nM) for 24 h. The protein expression of LC3 was then assessed by Western blotting. (**C**) B16F1/GFP-TFEB cells were treated with either nalfurafine hydrochloride (Nalf, 10, 100 μM) or Torin1 (0.25 μM for 1 h). The cells were fixed for fluorescence imaging, and the nuclear localization of GFP-TFEB was analyzed. (**D**) B16F1 cells were treated with either nalfurafine hydrochloride (Nalf 10, 100 μM) or Torin1 (0.25 μM for 1 h). Protein expression was assessed by Western blotting using the indicated antibodies. (**E**) B16F1 cells pre-treated with α-MSH for 48 h were further incubated with nalfurafine hydrochloride (Nalf, 100 μM) for 24 h. Protein expression was assessed by Western blotting using the indicated antibodies. Additionally, the protein levels were measured by densitometry analysis. (n = 3, ns: non-significant, * *p* < 0.05, *** *p* < 0.001.) The scale bar indicates 10 μm.

**Figure 2 cells-12-00146-f002:**
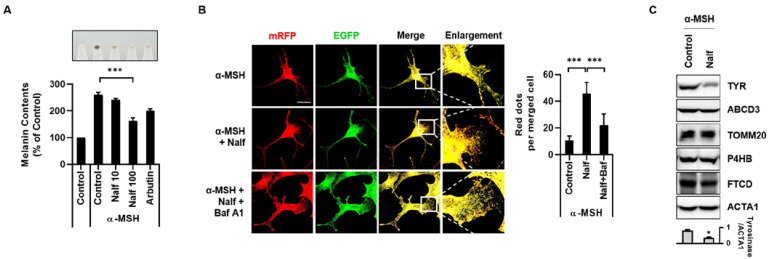
Nalfurafine hydrochloride decreases melanin content by enhancing the degradation of melanosomes in α-MSH-treated B16F1 cells. (**A**) B16F1 cells pre-treated with α-MSH for 48 h were further exposed to nalfurafine hydrochloride (Nalf, 10, 100 μM) or Arbutin (500 μM) for an additional 24 h. Subsequently, the cells were harvested (cell pellets: upper panel), and melanin content was measured as described in the Materials and Methods. (**B**) B16F1/TPC2-mRFP-EGFP cells were pre-treated with α-MSH (0.5 μM) for 48 h and then further incubated with nalfurafine hydrochloride (Nalf, 100 μM) with or without bafilomycin A1 (Baf A1, 100 nM) for 24 h. The cells were fixed, and the distribution of TPC2-mRFP-EGFP was imaged with confocal microscopy. The number of RFP-only puncta per cells was quantified from merged images. (**C**) B16F1 cells were pre-treated with α-MSH (0.5 μM) for 48 h and then further incubated with nalfurafine hydrochloride (Nalf, 100 μM). Cells were harvested and analyzed by Western blotting with the indicated antibodies (TYR: tyrosinase (melanosome), ABCD3 (peroxisome), TOMM20 (mitochondria), P4HB (endoplasmic reticulum), FTCD (Golgi). Additionally, the protein levels were measured by densitometry analysis. (n = 3, * *p* < 0.05, *** *p* < 0.001.) The scale bar indicates 10 μm.

**Figure 3 cells-12-00146-f003:**
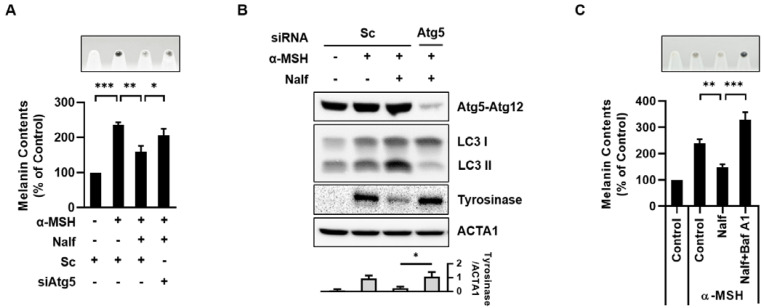
Depletion of ATG5 inhibits autophagy and restores melanin content in nalfurafine hydrochloride-treated B16F1 cells. (**A**,**B**) B16F1 cells were transfected with scrambled control siRNA (Sc) or Atg5 siRNA (siAtg5). After one day, the cells were further exposed to α-MSH (0.5 μM) for 48 h and treated with nalfurafine hydrochloride (Nalf, 100 μM) for an additional 24 h. After 72 h transfection of siRNAs, the cells were harvested. (**A**) The cell pellets were visualized (upper panel), and melanin content was measured. (**B**) Protein expression was assessed by Western blotting using the indicated antibodies. (**C**) B16F1 cells were treated with nalfurafine hydrochloride (Nalf, 100 μM) in the presence or absence of bafilomycin A1 (Baf A1, 100 nM) for 24 h. The cell pellets were harvested and visualized (upper panel), and the melanin content was measured. The protein levels were measured by densitometry analysis. (n = 3, * *p* < 0.05, ** *p* < 0.01, *** *p* < 0.001).

**Figure 4 cells-12-00146-f004:**
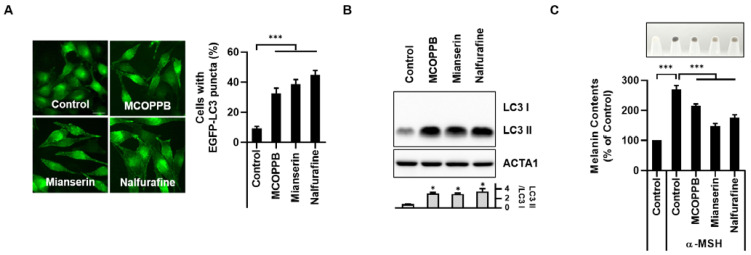
Treatment of KOR agonists, MCOPPB and Mianserin, induces melanophagy in B16F1 cells. (**A**) B16F1/GFP-LC3 cells were treated with MCOPPB (10 μM), mianserin (50 μM), or nalfurafine hydrochloride (Nalf, 100 μM) for 24 h. (**A**) After 24 h treatment, the cells were fixed for fluorescence imaging. The number of cells with activated autophagy activation was determined by counting punctate GFP-LC3 dots under a fluorescence microscope. (**B**) B16F1 cells were treated with MCOPPB (10 μM), mianserin (50 μM), or nalfurafine hydrochloride (100 μM) for 24 h. The level of LC3 protein was then assessed by Western blotting. (**C**) B16F1 cells pre-treated with α-MSH (0.5 μM) for 48 h were further treated with MCOPPB (10 μM), mianserin (50 μM), and nalfurafine hydrochloride (100 μM) for an additional 24 h. After harvesting, the cell pellets were shown (upper panel), and the melanin content was measured, as described in the Materials and Methods. (n = 3, * *p* < 0.05, *** *p* < 0.001.) The protein levels were measured by densitometry analysis. The scale bar = 10 μm.

**Figure 5 cells-12-00146-f005:**
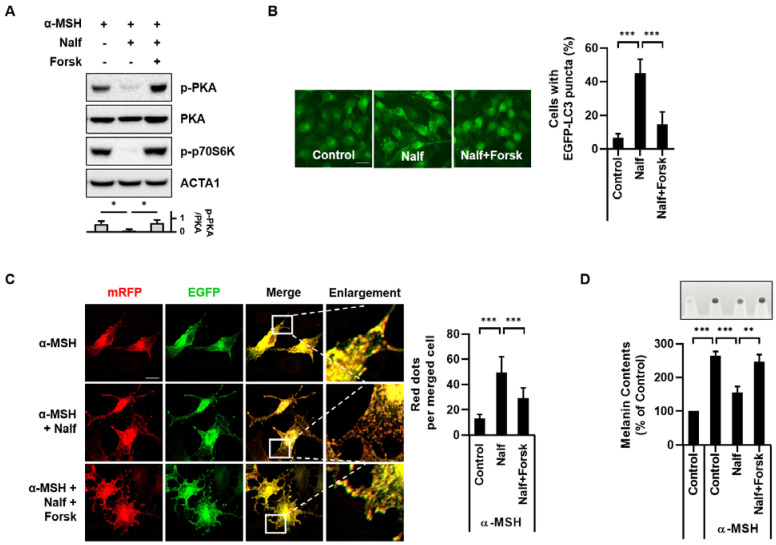
Inhibition of PKA by nalfurafine hydrochloride induces melanophagy in B16F1 cells. (**A**) B16F1 cells were pre-treated with α-MSH for 48 h and then further incubated with nalfurafine hydrochloride (Nalf, 100 μM) for 24 h in the presence or absence of forskolin (Forsk, 20 μM) for 4 h. Subsequently, cell lysates were harvested for Western blotting. Additionally, the protein levels were measured by densitometry analysis. (**B**) B16F1/GFP-LC3 cells were treated with nalfurafine hydrochloride (Nalf, 100 μM) for 24 h in the presence or absence of forskolin (Forsk, 20 μM) for 4 h. Cells were then imaged with a fluorescence microscope. Cells with autophagy activation were addressed with GFP-LC3 puncta cells. (**C**) B16F1/TPC2-mRFP-EGFP cells pre-treated with α-MSH for 48 h were further incubated with nalfurafine hydrochloride (Nalf, 100 μM) for 24 h in the presence or absence of forskolin (Forsk, 20 μM) for 4 h. Subsequently, the cells were fixed, and the distribution of melanosomes with TPC2-mRFP-EGFP was imaged with confocal microscopy. Cells presenting RFP-only puncta were quantified with cell images. (**D**) B16F1 cells pre-treated with α-MSH for 48 h were further incubated with nalfurafine hydrochloride (Nalf, 100 μM) for 24 h in the presence or absence of forskolin (Forsk, 20 μM) for 4 h. The cells were then harvested, and the melanin content was measured. (n = 3, * *p* < 0.05, ** *p* < 0.01, *** *p* < 0.001) The scale bar indicates 10 μm.

## Data Availability

Not applicable.

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
