# Peer review of "Nalfurafine Hydrochloride, a κ-Opioid Receptor Agonist, Induces Melanophagy via PKA Inhibition in B16F1 Cells"

_cells, 2022, doi:10.3390/cells12010146_

Round 1

Reviewer 1 Report

The manuscript demonstrated that κ-opioid receptor agonist nalfurafine hydrochloride inhibit melanin production by inducing melanophagy, which is dependent on PKA inhibition. It is interesting and helpful to researchers in the same field. However, it can not be accepted before some revision. My suggestions are listed below.

1.     Since Baf A1 is an inhibitor of autophagy, it is recommended to analyze its effects on nalfurafine hydrochloride treatment when used in combination. If Baf A1 reversed the effect of nalfurafine hydrochloride on melanin production, the conclusion will be more solid.

2.     The author used other opioid receptor agonists and found similar results. Why did they use antagonist of opioid receptor?

3.     How many times has the authors done their western blot experiments? The experiments should be done at least three times and provided the statistical results.

Author Response

Please Find an attached file

Reviewer 2 Report

Manuscript entitled "Nalfurafine hydrochloride, a κ-opioid receptor agonist, induces melanophagy via PKA inhibition in B16F1 cells" describes the action of Nalfurafine in melanophagy pathway. Although the authors followed the protocols to prove the hypothesis, the manuscript is still not sufficiently comprehensive for a research paper and a major revision is required. I have the following concerns/comments:  

1. Authors performed a cell-based high-content screening with an endocrinology-hormone library and identified nalfurafine hydrochloride as a potent inducer of autophagy in B16F1 cells. Which were the hormones analysed? How did nalfurafine identify to have an effect? What about the other compounds?

2. What was the rationale for the selection of doses 10 and 100μM nalfurafine? 

3. What was the significance of targeting mTOR using Torin1?

4. Results section description can be improved 

5.  Nalfurafine hydrochloride is a selective kappa (κ)-opioid receptor agonist. But mianserin is a tetracyclic antidepressant with multiple mechanism of action in histamine, serotonin and norepinephrine. Then how to justify the use of mianserin to explain the activation of κ-opioid receptors  to induce melanophagy in B16F1 cells.

6. Results section - Activation of the κ-opioid receptor inhibits the cyclic adenosine monophosphate (cAMP)/protein kinase A (PKA) signaling pathway [31].Yamamizu, K.; Furuta, S.; Hamada, Y.; Yamashita, A.; Kuzumaki, N.; Narita, M.; Doi, K.; Katayama, S.; Nagase, H.; Yamashita, 491 J.K.; et al. к Opioids inhibit tumor angiogenesis by suppressing VEGF signaling. Sci Rep 2013, 3, 3213, doi:10.1038/srep03213. But does this reference has evidence on cAMP levels? Give more evidences on the inhibition of cAMP/PKA by k-opioid receptor. 

7. How does a Nalfurafine acting through GPCR acts on mTOR signalling  and cAMP pathway. Explain the connection between both. Whether cAMP controls mTOR?

8. Discussion section requires further improvement as it repeats the results. It should link all signalling mechanisms. For instance there is no proper discussion the translocation of TFEB.

Author Response

Pleasee Find an attached file

Round 2

Reviewer 1 Report

They have not answered my questions adequately. I think they should add some experiments using antagonists. But they only said that they have used agonists. Besides, they only provided some of the statistic results of wb experiments. 

Author Response

Q1. They have not answered my questions adequately. I think they should add some experiments using antagonists. But they only said that they have used agonists.

Response 1:  During the first revision, we did not understand the question. Previously, it was reported that naltrexone a pan opioid receptor antagonist partially suppresses morphine-mediated autophagy by regulating endoplasmic reticulum stress in primary neurons (doi.org/10.1083/jcb.201605065). We had also examined an antagonistic effect of opioid receptor with naltrexone on melanophagy in B16F1 cells. However, we found that naltrexone has no proper effect on inhibition of melanophay.

Q2. Besides, they only provided some of the statistic results of wb experiments. 

Response 2: According to the suggestion, we added graphs with statistical analysis in the Figures.

Reviewer 2 Report

The authors have satisfactorily addressed most of my concerns.

Author Response

Thanks for the valuable comments